# Prognostic value of cervical length for spontaneous preterm birth in asymptomatic women with singleton pregnancy: An individual participant data meta-analysis

Kelly Hughes[1]*, David Nguyen[2], Mason Aberoumand[2], Heather Ford[3], Erin Clarke[1], Nuria Banos Lopez[4], Margaret Dziadosz[5,6], Richard Fischer[7], Renato T. Souza[8], Jose Guilherme Cecatti[8], Kelly Orzechowski[9], Courtney Olson-Chen[10], Alberto Borges Peixoto[11], Vorapong Phupong[12], Joshua Rosenbloom[13], Moeun Son[14], Athena Souka[15], Liu Du[16], Michael Sean Esplin[17], Roberta Granese[18], Simi Gupta[19], Brenda Kazemier[20,21], Lindsay Kindinger[22], Pihla Kuusela[23], Jeanine Van der Ven[24], Omer Weitzner[25,26], Evelyn Minis[26,27], Alba Farras Llobet[28], Heather Frey[29], Rashmi Bagga[30], Siddhidatri Mishra[30], Elizabeth Patberg[31], Philip Bennett[32], Megan Hall[33], Andrew Shennan[33], Shaun Brennecke[34,35], Shakila Thangaratinam[36,37,38], Anna Lene Seidler[39,40], Ben Willem Mol[1,41], Rui Wang[2]

1 Department of Obstetrics and Gynaecology, Monash University, Melbourne, Australia, 2 NHMRC Clinical Trials Centre, Faculty of Medicine and Health, University of Sydney, Sydney, Australia, 3 Pauline Gandel Imaging Centre, Royal Women's Hospital, Melbourne, Australia, 4 Department of Maternal-Fetal Medicine, BCNatal, Barcelona Centre for Maternal-Fetal and Neonatal Medicine, Hospital Clínic and Hospital Sant Joan de Déu, University of Barcelona, Barcelona, Spain, 5 RWJ Barnabas Health, Jersey City, New Jersey, United States of America, 6 New Jersey Perinatal Associates, Livingston, New Jersey, United States of America, 7 Department of Obstetrics and Gynecology, Cooper Medical School of Rowan University, Cooper University Hospital, Camden, New Jersey, United States of America, 8 Department of Obstetrics and Gynecology, Faculty of Medical Sciences, University of Campinas, Sao Paulo, Brazil, 9 Obstetrics and Gynecology Academic Department, School of Medicine, Georgetown University, Washington, District of Columbia, United States of America, 10 Department of Obstetrics and Gynecology, University of Rochester Medical Center, Rochester, New York, United States of America, 11 Department of Obstetrics and Gynecology, University of Uberaba, Minas Gerais, Brazil, 12 Department of Obstetrics and Gynecology, Chulalongkorn University, Bangkok, Thailand, 13 Department of Obstetrics and Gynecology, The Hebrew University of Jerusalem, Jerusalem, Israel, 14 Department of Obstetrics and Gynecology, Weill Cornell Medical College, New York, New York, United States of America, 15 Department of Obstetrics and Gynecology, National and Kapodistrian University of Athens School of Medicine, Athens, Greece, 16 Department of Medical Ultrasonics at The First Affiliated Hospital of Sun Yat-sen University, Guangzhou, Guangdong, China, 17 Department of Obstetrics and Gynecology, University of Utah, Salt Lake City, Utah, United States of America, 18 Department of Biomedical, Dental, Morphological and Functional Imaging Science, University of Messina, Messina, Italy, 19 Carnegie Imaging for Women, Mount Sinai West, New York, New York, United States of America, 20 Department of Obstetrics and Gynecology, Amsterdam UMC, Amsterdam, The Netherlands, 21 Amsterdam Reproduction and Development Research Institute, Amsterdam, The Netherlands, 22 Obstetrics and Gynaecology, University of Western Australia Medical School, Perth, Western Australia, Australia, 23 Research Department of Obstetrics and Gynecology, University of Gothenburg, Göteborg, Sweden, 24 Department of Obstetrics and Gynecology, Meir Medical Center, Kfar Saba, Israel, 25 School of Medicine, Faculty of Medical and Health Sciences, Tel Aviv University, Tel Aviv, Israel, 26 Division of Reproductive Endocrinology and Infertility, Department of Obstetrics and Gynecology, Massachusetts General Hospital, Boston, Massachusetts, United States of America, 27 Department of Obstetrics, Gynecology, and Reproductive Biology, Harvard Medical School, Boston, Massachusetts, United States of America, 28 Maternal and Fetal Medicine, Vall d'Hebron Research Institute, Barcelona, Spain, 29 Department of Obstetrics and Gynecology, The Ohio State University Wexner Medical Center, Columbus, Ohio, United States of America, 30 Department of Obstetrics and Gynaecology, Post Graduate Institute of Medical Education and Research, Chandigarh, India, 31 Department of Obstetrics, Gynecology and Reproductive Services, University of California San Francisco, San Francisco, California, United States of America, 32 Department of Metabolism, Digestion and Reproduction, Imperial College London, London, United Kingdom, 33 Division of Women's Health, King's College London, Women's Health Academic Centre, St

**Data availability statement:** The data included in this study are subject to restrictions by the Human Research Ethics Committee (HREC) and require data use agreements for access, therefore they are not publicly available. For access to the dataset, please contact the HREC of the University of Sydney (human.ethics@sydney.edu) and refer to the relevant authors listed in Table B in S1 Appendix. The statistical code is provided in the Supporting information.

**Funding:** This project was funded by the Australian National Health and Medical Research Council. BWM is supported by a NHMRC Practitioner Fellowship (GNT1082548). BWM and RW received a NHMRC Project Grant (GNT1146590) which also provided some support to KH. The funder had no role in study design, data collection and analysis, decision to publish, or preparation of the manuscript.

**Competing interests:** I have read the journal's policy and the authors of this manuscript have the following competing interests: BWM reports consultancy for Merck KGaA and Guerbet. BWM is supported by a NHMRC Practitioner Fellowship (GNT1082548) and reports consultancy for Merck KGaA and Guerbet. BWM and RW received a NHMRC Project Grant (2017 APP1146590) which also provided some support to KH. MS received research funds paid to Yale University from Sera Prognostics, Inc (https://www.sera.com/) for a separate project, the PRIME study. AS is an Academic Editor on PLOS Medicine's editorial board. The other authors report no conflict of interest.

**Abbreviations:** BMI, body mass index; CINAHL, cumulative index to nursing and allied health lIterature; CIs, confidence intervals; HR, hazard ratio; IPD, individual participant data; IPDMA, individual participant data meta-analysis; JBI, johanna briggs institute; LILACS, latin America and the caribbean literature on health sciences database; ORs, odds ratios; PPROM, pre prelabor rupture of membranes; PrI, prediction interval; PRISMA, preferred reporting Items for systematic reviews and meta-analyses; QUIPS, QUality In Prognosis Studies; SD, standard deviation; SPTB, spontaneous preterm birth.

Thomas' Hospital, London, United Kingdom, **34** Pregnancy Research Centre, Royal Women's Hospital, Melbourne, Australia, **35** Department of Obstetrics, Gynaecology and Newborn Health, The University of Melbourne, Melbourne, Australia, **36** Institute of Life Course and Medical Sciences, University of Liverpool, Liverpool, United Kingdom, **37** Liverpool Women's NHS Foundation Trust, Liverpool, United Kingdom, **38** NIHR North West Coast Applied Research Collaboration, University of Liverpool, Liverpool, United Kingdom, **39** Department of Child and Adolescent Psychiatry, University Medical Centre Rostock, Rostock, Germany, **40** German Center for Child and Adolescent Health (DZKJ), Partner Site Greifswald/Rostock, Rostock, Germany, **41** Department of Obstetrics and Gynaecology, University of Aberdeen, Aberdeen, Scotland, United Kingdom

* kelly.hughes@monash.edu

# Abstract

## Background

Spontaneous preterm birth (SPTB) is the leading cause of perinatal and early childhood mortality worldwide. Studies have generally suggested that mid-trimester transvaginal sonographic cervical length <25 mm is an important predictor of SPTB. Aggregate data meta-analyses are limited by data availability and reporting in the primary literature. The purpose of this individual participant data meta-analysis (IPDMA) was to quantify the prognostic value of mid-trimester cervical length for SPTB in asymptomatic women with singleton pregnancy, and to assess other factors which may modify this association.

## Methods and findings

The project was prospectively registered with PROSPERO (CRD42020146987). We searched Medline, Embase, CINAHL, LILACS, Database of Abstracts of Reviews of Effects (DARE), Cochrane database, JBI Database of Systematic Reviews, ClinicalTrials.gov, and Google Scholar. We included cohort studies and non-treatment arms of randomized controlled trials which assessed an association between mid-trimester transvaginal sonographic cervical length and SPTB in asymptomatic women with singleton pregnancy. The search was performed on 30/9/2020, with an update performed on 4/11/2025. The primary outcome was STPB <37 weeks. Two reviewers screened all studies for inclusion and performed risk of bias assessments using QUIPS. We performed a two-stage IPDMA in a logistic regression model using cervical length as a continuous variable (the primary analysis) with restricted cubic splines to explore non-linear associations.

IPD of 27 eligible studies were obtained and included ($n$ = 91,404). Mean cervical length was 40 mm (standard deviation [SD] 9 mm) at about 20 weeks' gestation. SPTB <37 weeks occurred in 4,442 (5.2%) participants. An L-shape non-linear association between cervical length and SPTB was observed. A longer cervical length was associated with steeply lower odds of SPTB until it reached 40 mm, beyond which the odds of SPTB became stable. This means that compared to a woman with a cervical

length of 40 mm, those with a cervical length of 20 and 30 mm were associated 6.22 and 2.10 higher odds of SPTB (95% confidence intervals [4.76, 8.13] and [1.85, 2.38]), respectively. Limitations included suboptimal data retrieval rate (51% of all eligible participants) and a lack of comprehensive co-predictors of SPTB across all datasets.

## Conclusion

We found a non-linear association between cervical length and SPTB. We found a non-linear association between cervical length and SPTB. Shorter cervix is associated with progressively higher risk of SPTB when length is less than 40 mm, but probability of term birth is high when cervical length is over 40 mm.

Author summary

**Why was this study done?**

- Preterm birth (before 37 weeks of pregnancy) is the main cause of death and illness in infants around the world.

- There is a well-established relationship between a short cervix—measured in the second trimester of pregnancy by vaginal (internal) ultrasound—and subsequent preterm birth.

- There are important limitations in the many studies that have been performed in the past.

- Better prediction of preterm birth means more ability to offer treatments to reduce risk.

**What did the researchers do and find?**

- We collected 27 datasets from previously conducted studies including >90,000 singleton pregnancies to combine and re-analyse with up-to-date statistical techniques, using the measurement taken at the time of the routine morphology ultrasound at around 20 weeks' gestation.

- We confirmed and quantified the relationship between a shorter cervix and earlier birth: for cervical lengths below 40 mm, there is a steep increase in the risk of a spontaneous preterm and very preterm birth (before 34 weeks of pregnancy) with shorter cervix.

**What do these findings mean?**

- Measuring the length of the cervix at around 20 weeks in the pregnancy remains an effective predictor of preterm birth.

- In the future, these findings may be used together with knowledge of treatment effects to help women share in decision-making about their pregnancy care.

- There may be other factors that could be combined with the cervical length to better predict preterm birth, but the studies we had did not always collect this information.

## Introduction

Preterm birth (before 37 weeks' gestation) is the leading cause of perinatal and early childhood mortality worldwide. It increases the risk of death from other causes and is associated with multiple short and long-term morbidities in survivors [1]. Preterm birth may be iatrogenic due to maternal or fetal indication, or spontaneous. Spontaneous preterm birth

(SPTB) represents a potentially predictable and preventable condition, with numerous proposed aetiologies [2], and a range of historical and demographic factors shown to increase risk [3]. Addressing modifiable risk factors for SPTB has the potential to improve childhood morbidity and mortality, reduce the psychological toll on families, and provide substantial economic and societal cost savings.

Vaginal progesterone and cervical cerclage are evidence-based treatments to reduce the incidence of SPTB in high-risk women [4,5]. Identification of women at high risk can allow targeted use of these prophylactic measures, however, predicting SPTB has proved to be challenging despite knowledge of various risk factors [6].

Cervical length measured via transvaginal ultrasound in the second trimester in asymptomatic women is acknowledged as an important predictor for SPTB [7]. We have previously described the important limitations in existing aggregate data meta-analyses which restrict their applicability [8]. Key issues included: insufficient reporting often meant aggregate data was unavailable; arbitrary dichotomisation in primary studies using a wide range of cut-offs for the prognostic factors and primary outcome limited the ability to synthesise aggregate data. We also questioned the validity of the current practice of dichotomising cervical length to assign risk: a woman with a 24 mm cervical length being at high risk of SPTB, yet a woman with a 26 mm cervical length would be considered normal risk [8], inter-rater reliability notwithstanding [9]. Additionally, previous meta-analyses have used diagnostic test methodology rather than a contemporary prognostic factor approach [8].

Performing an individual participant data meta-analysis (IPDMA), collating row-by-row raw datasets, is the "platinum standard" [10] for overcoming these limitations. It allows inclusion of all available data, application of consistent definitions of outcomes, and analysis of variables on a continuous scale [11]. The primary objective of this IPDMA was to quantify the prognostic value of mid-trimester sonographic cervical length for SPTB in asymptomatic women with singleton pregnancy. We also aimed to assess this relationship in women with and without specific risk factors for SPTB, whether other co-factors would enhance the prognostic value of cervical length, and to assess any relationship with neonatal outcomes.

## Methods

### Protocol and registration

The research protocol was registered with PROSPERO, CRD42020146987. We have previously published the overview of systematic reviews, whose literature search underpins this IPDMA [8]. The manuscript was reported according to PRISMA-IPD guideline. The project was granted ethics approval by both The University of Sydney (Project no. 2019/397) from 14th May 2019 and Monash University from 27th April 2021 (Project no. 27571).

### Eligibility criteria

Studies were eligible for inclusion if meeting the following criteria:

- Participants: asymptomatic women with singleton pregnancy, with or without additional risk factors for SPTB, where not more than 20% of cohort received treatment to reduce risk of SPTB (progesterone, cerclage, pessary), $n \geq 100$.

- Indexed prognostic factor: transvaginal ultrasound performed with a widely accepted technique (i.e., as described by Iams and colleagues [7] or Fetal Medicine Foundation [12]).

- Outcome: when both gestational age at delivery and classification as spontaneous or iatrogenic preterm birth are available in individual participant data (IPD).

- Timing: pregnancy dated by standard methods (early dating ultrasound or last menstrual period), gestational age at transvaginal cervical length measurement between 16 and 26 weeks' gestation, data collected after the year 2000.

- Design: cohort studies, or placebo/untreated groups from randomised controlled trials.

## Identifying studies—Information sources

An information specialist helped design the search strategy; the full details are provided in Table A in S1 Appendix. We searched Medline, Embase, CINAHL, LILACS, Database of Abstracts of Reviews of Effects (DARE), Cochrane database, JBI Database of Systematic Reviews, ClinicalTrials.gov, and Google Scholar. The search was restricted to 1st January 2000 and beyond due to anticipated limited availability of older data. No language restriction was applied. We performed citation tracking and consulted experts in the field. The search was performed on 30th September 2020, with a post-analysis updated search performed on 4th November 2025.

## Identifying studies—Search

We searched the following keywords: cervix uteri, uterine cervical incompetence, cervical length measurement, ultrasonography, prenatal, (cervix or cervical) length, (pre-term or preterm or premature), (delivery or birth or labour or labor), abortion, spontaneous (Table A in S1 Appendix).

## Study selection processes

Title and abstract screening were performed by two authors (KH, RW) using Covidence systematic review software (Veritas Health Innovation, Melbourne, Australia). Full-text review was completed by two reviewers (KH, HF); any conflicts were resolved by a third reviewer (RW/BWM).

## Data collection processes

We sought contact details for the corresponding and/or first author, or, where that was unavailable or obsolete, any listed author. We sent an introductory email to at least one author of each citation, and at least two further emails (with alternate addresses tried where available) over the course of several months if no response was received. Where contact details were outdated, we attempted to locate the author's current institution via internet and PubMed searching or enquired via other research contacts in a similar geographical area.

If an author responded positively, we arranged an online meeting and/or a data-sharing agreement. De-identified data was transferred by secure file transfer platform or e-mail according to contributor preference and uploaded to a secure university server. Data queries were resolved by e-mail correspondence. IPD was sought for all eligible studies where an author was contactable, including those published only as conference abstracts.

## Data items

We requested entire de-identified datasets, at minimum including cervical length, gestational age at measurement, and gestational age at birth. Cervical length measured in millimetres nearest to 20 weeks' gestation was the prognostic factor of interest. In cases where only a range for gestational age at the time of measurement of the cervical length for the entire individual study was provided rather than individual-level data, the mean of that range was imputed as the gestational age. In cases where multiple cervical length measurements were available per participant, we selected the measurement closest to 20 weeks' gestation due to it coinciding with the usual morphology ultrasound.

We also requested other potential prognostic factors including but not limited to maternal age, BMI, nulliparity, history of preterm births, cervical surgery, tobacco smoking, and use of artificial reproductive technology. Prognostic factor selection among pre-specified clinically meaningful variables was guided by the availability of data across studies (at least 50% of data points) and descriptive analyses of association between factor and outcome.

The primary outcome was SPTB <37 weeks' gestation. Preterm prelabour rupture of membranes (PPROM) was counted as a SPTB even if delivery was due to induction of labour or if it occurred after 37 weeks; the rationale being that the pathological process commenced prematurely.

The secondary outcomes included SPTB <34 weeks and SPTB <30 weeks. We also presented the time to SPTB <37 weeks as a time-to-event outcome in a sensitivity analysis where term births were censored at 37 weeks. Where studies also reported medically indicated preterm birth, these were treated as censored observations.

## IPD checks

All data was checked with respect to range, internal consistency, missing or extreme values, errors, and consistency with published reports prior to conducting the analyses. Concerns about data inconsistency and integrity were solved by contacting the leading author of the original publication. If concerns could not be addressed adequately, we excluded the dataset in question.

## Risk of bias assessment in individual studies

Risk of bias was assessed in six domains using the QUality In Prognosis Studies (QUIPS) tool [13]; this included study participation, study attrition, prognostic factor measurement, outcome measurement, adjustment for other prognostic factors, and statistical analysis and reporting. For domains on adjustment for other prognostic factors and statistical analysis and reporting, the assessments were based on the IPD. Two reviewers (KH, EC) assessed the included studies, conflicts were resolved by consensus or a third reviewer (RW). Summary figures for risk of bias were created with robvis [14].

## Synthesis methods

Datasets were compiled with SAS software version 9.4 [15], harmonising and analysis were performed with R [16] and Stata [17]. We performed a two-stage IPDMA because most included studies were of adequate sample size. The first stage comprised building regression models for each study, using all the explanatory variables available in each study. We built logistic regression models, producing odds ratios (ORs) with 95% confidence intervals (CIs). We treated cervical length as a continuous variable and used restricted cubic splines to investigate the non-linear relationship between cervical length and SPTB for each trial. Four knots were selected using the centiles recommended by Harrell [18] (at 5th, 35th, 65th, and 95th centiles) based on the overall dataset including all studies. The same values (instead of centiles) of the four knots were then applied to each study.

We built multiple models for the primary outcome, as follows:

1. Basic model including only cervical length and gestational age at measurement as prognostic factors (Model 1)—primary model

2. Model with only key demographic factors (mother's age, nulliparity) and cervical length (Model 2)

3. Model with all potential prognostic factors (Model 3a), and model with all potential prognostic factors, only including studies that include all factors (Model 3b)

In the second stage, the results of the study-level regression models were meta-analysed in a random-effects model using the inverse variance method. For linear associations, forest plots were produced for each model, and the percentage of the total variability in study estimates due to between-study variance was determined using $I^2$ statistic. Between-study variance was measured with $\tau^2$ and the 95% prediction interval (PrI) was calculated using a previously described inverse variance method [19]. For non-linear associations, multivariate random-effects meta-analyses using restricted maximum likelihood were performed. The associations were visualised in restricted cubic splines to illustrate the odds of SPTB at different cervical lengths compared to those at a reference cervical length (40 mm). Tables with ORs and 95% CIs were also produced to assist with the interpretation.

Where data were systematically missing at study level, the two-stage meta-analysis described here allowed for outcome estimates using all the available variables in each study. Complete case analysis was performed.

The following pre-specified and post-hoc subgroup analyses (exposure–covariate interaction) were performed when data were available, and only within-study interaction was considered:

- History of excisional cervical surgery (large loop excision of the transformation zone/loop electrosurgical excision, cone biopsy)

- Uterine anomaly

- Previous preterm birth (post-hoc)

- Previous term birth (post-hoc)

- Nulliparous women (post-hoc)

- Gestational age at measurement (post-hoc)

The following sensitivity analyses for the primary model were performed when the data were available:

- Exclude women who receive treatment to reduce the risk of preterm birth (cerclage, progesterone, pessary).

- Exclusion of studies with an overall high risk of bias.

- Time to SPTB (<37 weeks), time-to-event outcome. We built Cox proportional hazard models, producing hazard ratios (HRs) with 95% CIs. Gestational age at measurement was used as the entry time variable, and GA at delivery as the exit time variable.

We also performed the following post-hoc sensitivity analyses suggested by the reviewers:

- Restricting to participants with gestational age of measurement between 18 and 21 + 6 weeks.

- For time to SPTB (<37 weeks), competing risk model was used, and iatrogenic preterm birth was considered as a competing event to SPTB.

### Deviations from the statistical analysis plan

Following the advice from peer reviewers, we used logistic regression, instead of Cox regression, as the primary analysis and treated the primary outcome as a binary outcome to make the paper more accessible to all readers. We still present the Cox regression on time to SPTB findings in sensitivity analyses (Figs D1-6 in S1 Appendix).

## Results

### Search results

The initial search yielded 6,085 citations. After removal of duplicates and full-text screening, IPD was sought for 109 eligible citations, however, for 64 of them, authors were unable to be contacted or did not complete the data sharing process for various reasons. We obtained IPD from 27 datasets, comprising 91,404 pregnancies from an estimated possible 183,903 (50.7% of IPD) as outlined in the PRISMA flow diagram (Fig 1). One dataset was subsequently excluded due to data integrity concerns ($n = 200$). One dataset ($n = 7,954$) was excluded due to cervical length measurements only available for 6% of participants. Studies where IPD could not be obtained, and the reasons for unavailability are listed in Table B in S1 Appendix.

At times, two or more citations (denoted "records" in Fig 1) were found to have been produced from a single dataset (denoted "study" in Fig 1). Where authors were unable to be contacted, we could not verify whether one record corresponded to one study, but this was assumed to be the case for recording the number of pregnancies that could not be included in our combined dataset.

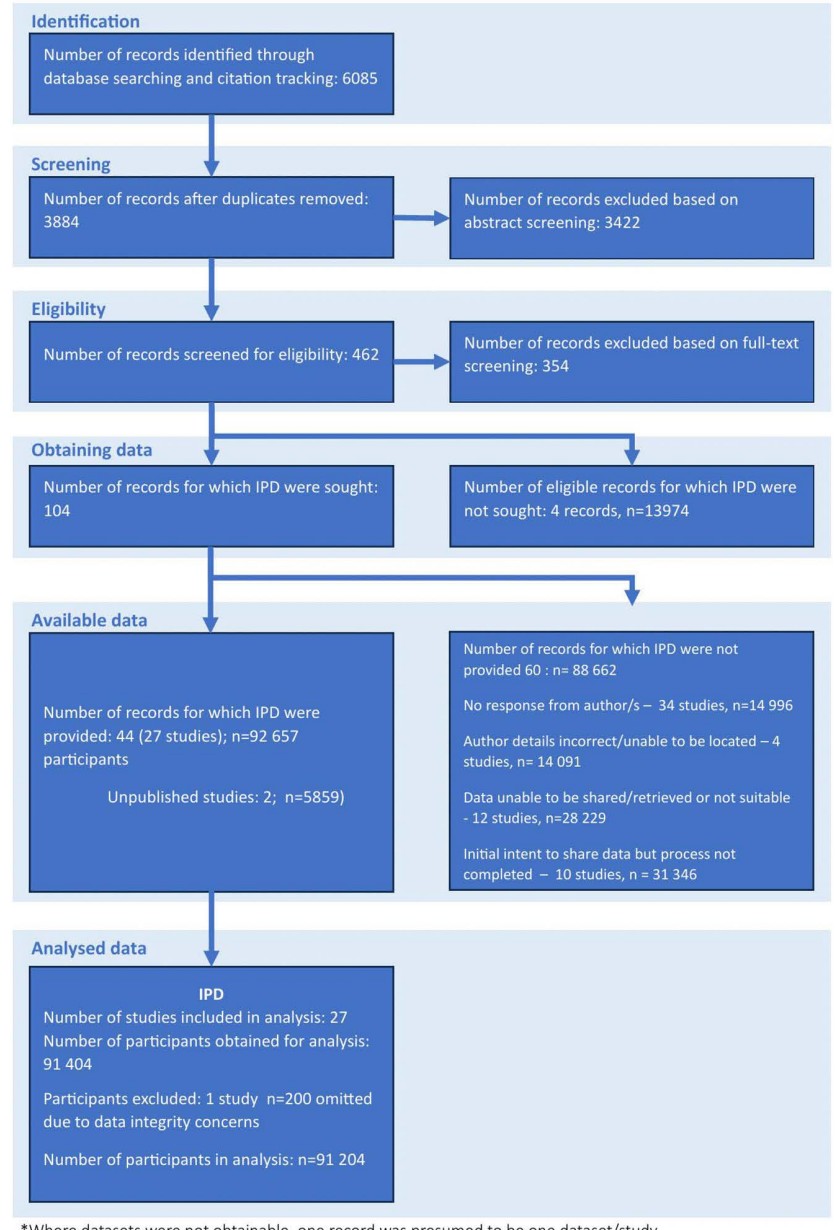

**Fig 1. PRISMA flow diagram.**

## Study characteristics

Study characteristics and demographic data are shown in Tables 1 and 2. The studies were conducted in 12 countries between 2001 and 2019. Fifty-five% of the study population were nulliparous (*n* = 46,195). Mean gestational age at

**Table 1. Included studies.**

| Study | Year | Country | Study design | Population risk level | Number of participants | Mean CL in mm (±SD) | Mean GA at measurement in weeks (±SD) | Mean GA at delivery in weeks (±SD) | SPTB <37 weeks, n (%) | SPTB <34 weeks, n (%) |
|---|---|---|---|---|---|---|---|---|---|---|
| Banos Lopez [20] | 2018 | Spain | Prospective cohort | Normal | 842 | 39.4 (6.3) | 21.1 (1.2) | 39.2 (2.1) | 53/818 (6.5) | 15/818 (1.8) |
| Bennett | Institution-based dataset | UK | Prospective cohort | High | 3,941 | 32.4 (8.3) | 19.7 (2.7) | 37.3 (4.8) | 270/2553 (10.6) | 160/2800 (5.7) |
| Du [21] | 2020 | China | Prospective nested case-control | Normal | 286 | 36.4 (5.7) | 21.5 (1.5) | 38.8 (1.6) | 26/286 (9.1) | 7/286 (2.4) |
| Dziadosz [22] | 2016 | USA | Retrospective cohort | Unselected | 1,097 | 20.4 (4.1) | 20.4 (4.1) | 38.9 (2.6) | 72/1097 (8.6) | 31/1097 () |
| Esplin [23] | 2017 | USA | Prospective cohort | Unselected nulliparous | 9,289 | 39.1 (8.3) | 19.1 (1.5) | 38.9 (3.4) | 367/8776 (4.2) | 106 (1.2) |
| Farras Llobet [24] | 2020 | Spain | Prospective cohort | Unselected | 1,453 | 36.8 (5.7) | 20.5 (0.7) | 39.2 (1.9) | 52/1410 (3.7) | 2/1453 (0.14)) |
| Fischer [25] | 2010 | USA | Prospective cohort | High | 170 | 35.8 (7.2) | 19.2 (1.7) | 38.6 (2.5) | N/A | N/A |
| Gupta [26] | 2019 | USA | Retrospective cohort | High | 130 | 37.5 (5.8) | 21.1 (1.6) | 39.1 (1.9) | N/A | N/A |
| Kindinger [27] | 2016 | UK | Retrospective cohort | High | 727 | 32.0 (5.4) | N/A | 29.0 () | 73/725 (10.1) | 12/725 (1.66) |
| Kuusela [28] | 2015 | Sweden | Prospective cohort | Unselected | 2061 | 40.0 (6.4) | 18.6 (0.8) | 39.7 (1.8) | 87/2061 (4.2) | 22/2061 (1.1) |
| Minis, Witkin [29,30] | 2018, 2020 | Brazil | Prospective cohorts | Unselected | 1,250 | 33.2 (7.7) | 21.1 (1.2) | 38.6 (2.3) | 49/541 (9.1) | 10/559 (1.8) |
| Mishra [31] | 2018 | India | RCT | Normal | 150 | 34.4 (4.7) | 19.9 (2.3) | 38.4 (1.7) | 11/147 (7.5) | 2/147 (1.4) |
| Olson-Chen [32] | 2018 | USA | Prospective cohort | Normal and high-risk groups | 331 | 38.2 (7.5) | 18.9 (1.3) | 38.8 (2.6) | 29/331 (8.8) | 12/331 (3.6) |
| Orzechowski [33] | 2014 | USA | Prospective cohort | Unselected | 2,558 | 41.6 (7.3) | 20.2 (1.2) | 39.1 (2.2) | 88/1910 (4.6) | 29/2157 (1.3) |
| Patberg [34] | 2021 | USA | Retrospective cohort | Normal | 1,000 | 35.9 (6.0) | 20.6 (1.4) | 38.4 (2.3) | 75/937 (8.0) | 12/937 (1.3) |
| Peixoto [35] | 2017 | Brazil | Retrospective cohort | Unselected | 2,769 | 37 (6) | 22.3 (1.1) | 37.8 (3.5) | 235/1247 (18.9) | 110/1239 (8.9) |
| Pils [36] | 2014 | Austria | Retrospective cohort | High | 384 | 21.7 (3.6) | 21.7 (3.6) | 37.4 (4.0) | NA | NA |
| Puttanavijarn, Phupong [37] | 2017 | Thailand | Prospective cohort | Normal | 160 | 41.2 (5.3) | 18.8 (2.0) | 38.2 (1.4) | 14/160 (8.8) | 3/160 (1.9) |
| Rosenbloom [38] | 2020 | USA | Retrospective cohort | Normal | 13,508 | 41.0 (7.4) | 19.9 (1.2) | 38.6 (2.6) | 754/13508 (5.6) | 245/13508 (1.8) |
| Kuhrt, Shennan [39] | 2016 + institution-based dataset | UK | Prospective cohort | High | 1,918 | 35.9 (10.2) | 21.1 (3.4) | 38.1 (4.2) | 254/1886 (13.5) | 137/1916 (7.2) |
| Son [40] | 2016 | USA | Retrospective cohort | Normal | 17,590 | 45.0 (8.4) | 20.3 (0.9) | 39.3 (1.8) | 700/17589 (4.0) | 176/17587 (1.0) |
| Souka1 [41] | 2011 | Greece | Prospective cohort | Unselected | 3,390 | 31.2 (5.2) | 22.4 (0.7) | 38.8 (1.7) | 157/3357 (4.7) | 28/3337 (0.84) |

*(Continued)*

**Table 1.** (Continued)

| Study | Year | Country | Study design | Population risk level | Number of participants | Mean CL in mm (±SD) | Mean GA at measurement in weeks (±SD) | Mean GA at delivery in weeks (±SD) | SPTB <37 weeks, *n* (%) | SPTB <34 weeks, *n* (%) |
|---|---|---|---|---|---|---|---|---|---|---|
| Souka2 [42] | 2015 | Greece | Retrospective cohort | Unselected | 7,116 | 36.2 (6.4) | 22.1 (0.9) | 38.6 (1.5) | 253/7093 (3..6) | 56/7092 (0.79) |
| Souza, Cecatti [43] | 2020 | Brazil | Prospective cohort | High | 1,166 | 36.5 (6.8) | 21.7 (1.6) | 38.8 (2.4) | 78/1165 (6.7) | 23/1165 (2.0) |
| Van der Ven [44] | 2015 | Netherlands | Prospective cohort | Normal | 11,943 | 44.1 (7.8) | 20.3 (0.6) | 39.5 (1.8) | 464/11943 (3.9) | 102/11943 (0.85) |
| Ward, Frey [45] | 2019 | USA | Retrospective cohort | Unselected | 6,459 | 38.5 (9.9) | 19.3 (1.2) | 38.8 (2.3) | 267/6433 (4.6) | 104/6453 (1.6) |
| Weitzner [46] | 2019 | Israel | Retrospective cohort | Low | 198 | 38.3 (5.3) | n/a | 39.2 (1.7) | 14/194 (7.2) | 1/194 (0.5) |

CL, cervical length; SD, standard deviation; GA, gestational age; SPTB, spontaneous preterm birth; n/a, not applicable (where, after data transfer, iatrogenic preterm births unable to be identified).

UK, United Kingdom; USA, United States of America.

**Table 2.** Demographic data.

| Variable, category | Number of participants (total 91,404) (%)/mean (SD) or as specified |
|---|---|
| Maternal age (years) | 30.4 (5.6) |
| Missing | 5,512 (6) |
| Gravidity | 2.36 (1.67) |
| Missing | 61,173 (66.9) |
| Nulliparity | 37,029 (40.5) |
| Para ≥1 | 45,139 (49.4) |
| Missing | 6,330 (6.9) |
| Mode of conception | |
| Spontaneous | 18,156 (19.9) |
| IVF | 557 (0.6) |
| Missing/not specified | 69,785 (76.4) |
| Past obstetric history | |
| Previous preterm birth | 2,875 (3.1)55,823 (67.1) |
| Missing/not specified/no previous preterm birth | |
| Cervical length (mm) | 40 (9) |
| Missing | 8,495 (9.2) |
| Gestational age at cervical length measurement (weeks) | 20.32 (1.58) |
| Prophylactic treatment received (*n*, %) | |
| Cerclage | 1,216 (1.3) |
| Progesterone | 9,843 (10.8) |
| Pessary | 69 (0.07) |
| Gestational age at delivery (weeks) (mean, SD) | 38.9 (2.52) |
| SPTB <37 weeks (*n*, %) | 4,442 (5.2) |
| SPTB <34 weeks (*n*, %) | 1,420 (1.6) |

SD, standard deviation; SPTB, spontaneous preterm birth.

cervical length measurement was 20.3 weeks' gestation (standard deviation [SD] 1.6 weeks), and mean cervical length was 40 mm (SD 9 mm). Eleven,128 women (up to 12.2% of the cohort) received prophylactic treatment as shown in Table 2. Mean gestational age at birth was 38.9 weeks' gestation (SD 2.5 weeks); results by study are shown in Table 1. 4442 women (5.2%) had a SPTB <37 weeks' gestation, and 1,420 (1.6%) before 34 weeks. The rate of SPTB <37 weeks varied between studies (range 3.6%–18.8%). Cervical length and gestational age at delivery were weakly positively correlated (sample correlation estimate = 0.19, 95% CI [0.19.0.20]), see Fig A in S1 Appendix.

### Risk of bias assessments

Results for QUIPS risk of bias (RoB) assessment are shown in Fig 2A and 2B. 3/27 were rated as low RoB overall. 11/26 studies were assessed as being at high RoB overall. 19/26 studies were at moderate or high risk of bias in domain two (bias due to attrition).

### Individual participant data meta-analysis

The primary model (including only cervical length and gestational age at measurement, Model 1) showed an L-shape non-linear association between cervical length and SPTB (22 studies, 78,047 participants, Fig C1 in S1 Appendix Table C in S1 Appendix). We present the association on the odds scale for easier interpretation (Fig 3). A longer cervical length was associated with steeply lower odds of SPTB until it reached 40 mm, beyond which the odds of SPTB became stable with longer cervical length. Using cervical length of 40 mm as the reference, a woman with a cervical length of 20 mm was associated with 6.22 higher odds of SPTB (95% CI [4.76,8.13]) whereas a woman with a cervical length of 30 mm was associated with 2.10 higher odds of SPTB (95% CI [1.85,2,38]). The detailed summary statistics including the knots, coefficients, 95% confidence intervals, p values, and between-study heterogeneity (tau) for each restricted cubic spline term are presented in Table C in S1 Appendix.

Similar shapes of non-linear association between cervical length and SPTB were observed when adding maternal body mass index (BMI), maternal age and nulliparous status into the model (Model 2, see Figs B2, C in S1 Appendix), adding all potential prognostic factors (Model 3a, Fig 3 and Fig B3 in S1 Appendix), or model with all potential prognostic factors, only including studies that include all factors (Model 3b, Fig 3 and Fig B4 in S1 Appendix). For Models 2 and 3b, the ORs were larger than those in Model 1 when cervical length was short. As only 4 studies including 6,062 participants were included in Model 3b, the CIs for Model 3b are wide on both tails (Fig C4 in S1 Appendix). The detailed statistics for each model are presented in Table C in S1 Appendix.

The prognostic value of cervical length for SPTB <34 weeks was stronger, as shown with higher ORs at each cervical length value under 40 mm, compared to those on SPTB <37 weeks (Figs 4 and C1 in S1 Appendix). Above 40 mm, the ORs for SPTB <34 weeks did not increase with longer cervical length. Similarly, the prognostic value of cervical length for SPTB <30 weeks was even stronger (Figs 4 and C2 in S1 Appendix). By contrast, the prognostic value of cervical length on any preterm birth <37 weeks was weaker than those on SPTB < 37 weeks ( Figs 4 and 5 and C3 in S1 Appendix).

### Sensitivity analysis

Sensitivity analyses for SPTB <37 weeks excluding women who received cerclage, progesterone, or pessary, excluding studies with an overall high risk of bias or excluding participants with gestational age of cervical length measurement outside 18 to 21 + 6 weeks showed robust non-linear association curves, compared to the primary analysis (Fig D1–4 in S1 Appendix). The detailed statistics for all sensitivity analyses are presented in Table C in S1 Appendix.

A sensitivity analysis based on Cox regression models was also performed, with a similar shape of non-linear association curve (Fig D5 in S1 Appendix). When limiting to 10 studies reporting both spontaneous and iatrogenic preterm birth, another sensitivity on competing risk model was performed, which was consistent with the findings from the Cox regression (Fig D5 in S1 Appendix).

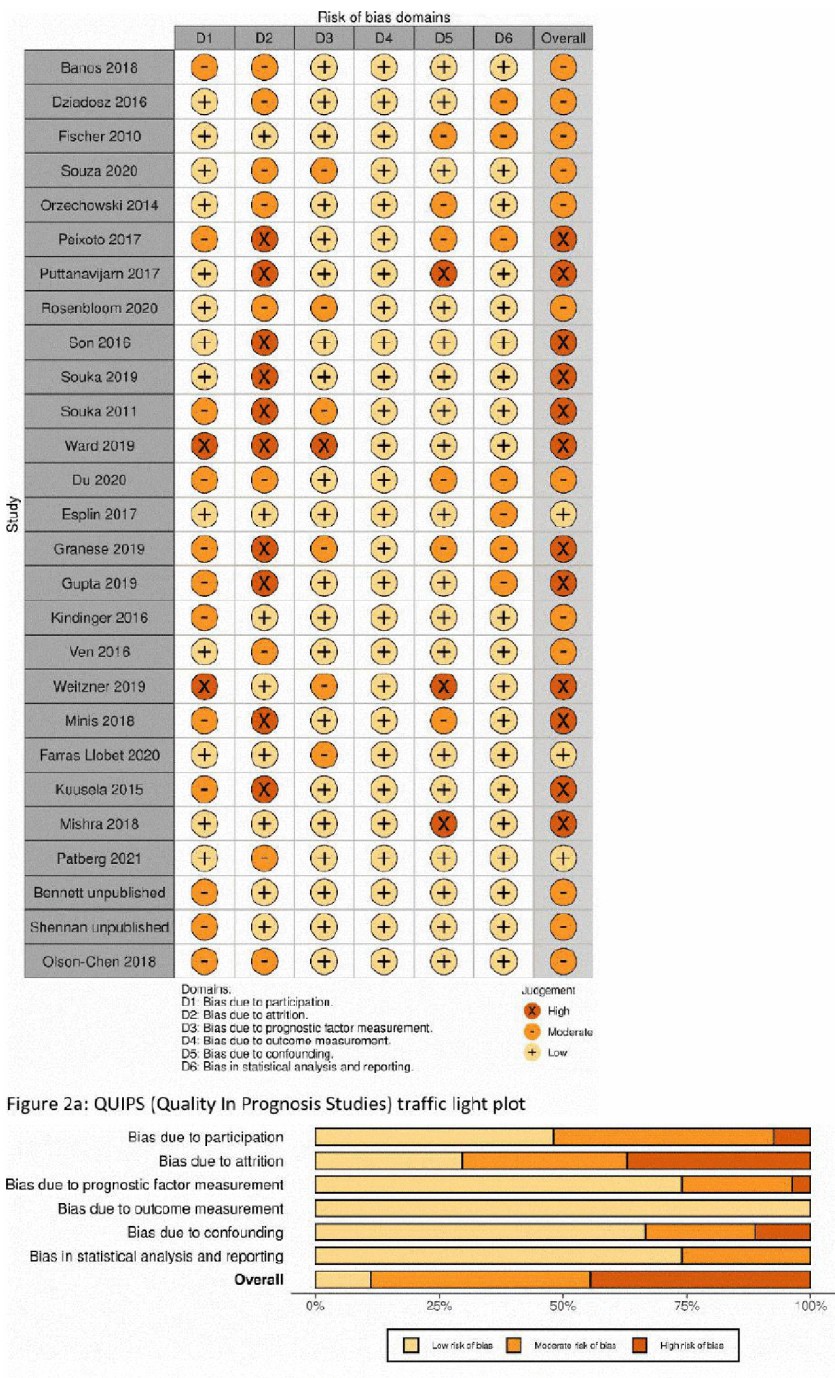

Figure 2a: QUIPS (Quality In Prognosis Studies) traffic light plot

Figure 2b: QUIPS (Quality In Prognosis Studies) summary plot

**Fig 2. Risk of bias assessment results. (A)** Quality In Prognosis Studies (QUIPS) traffic light plot. **(B)** QUIPS summary plot.

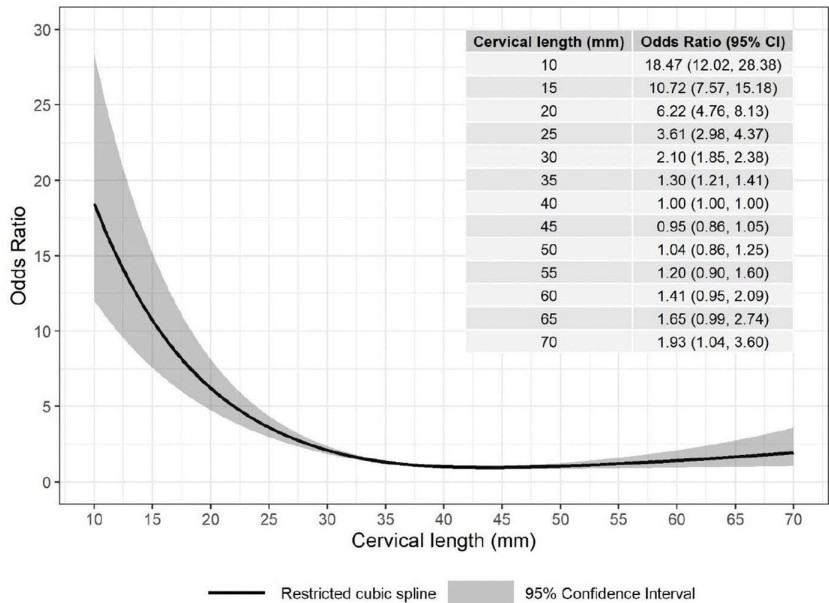

**Fig 3. Non-linear association between cervical length and spontaneous preterm birth <37 weeks in the primary model.** This figure illustrates ORs and 95% Cis of spontaneous preterm birth <37 weeks at different cervical length vs. the reference cervical length (40 mm) in the primary analysis (Model 1). Cervical length and gestational age at measurement were the only prognostic factors used in the model.

## Subgroup analysis (exposure–covariate interaction)

Planned subgroup analysis showed stronger prognostic value of cervical length on SPTB <37 weeks in women with a history of cervical surgery compared to women without a history of cervical surgery when cervical length was below 28 mm. Such a difference became inconclusive when cervical length was above 28 mm due to wider CIs (Fig E1 in S1 Appendix).

In women with a previous preterm birth, the prognostic value of cervical length on SPTB<37 weeks was weaker than those without such history when cervical length was below 25 mm. Such a difference became inconclusive when cervical length was above 25 mm due to wider CIs (Fig E3 in S1 Appendix).

No evidence of other potential interaction were observed between cervical length and(uterine anomaly, history of term birth, or nulliparity (Fig E2, E4, E5 in S1 Appendix). There was also no evidence of an interaction between gestational age at measurement and cervical length on SPTB <37 weeks (Fig D5 in S1 Appendix). The detailed statistics for all subgroup analyses are presented in Table C in S1 Appendix.

## Systematic review of studies identified during updated literature search

Following the literature search update on 4th November 2025, we identified 26 newly eligible studies (46,247 participants) for our IPDMA. In addition, six more studies may be eligible: they did not directly address the association between cervical length and SPTB, but may have information on gestational age, cervical length, and SPTB available in the datasets.

Study characteristics are presented in Table B in S1 Appendix. Of the 26 eligible studies, 11 (42%) were abstract-only reports, and 15 (58%) were full publications. Twelve (46%) had a prospective cohort study design, 14 (54%) had a retrospective cohort design. The median number of participants in the 26 eligible studies was 797 (interquartile range (IQR) 339–1,425), 13 (50%) reported the research question as a prognostic factor research question, and the other half were reported as a diagnostic test accuracy research question. Among the 13 studies addressing the association between cervical length and SPTB as a prognostic research question, 10 analysed

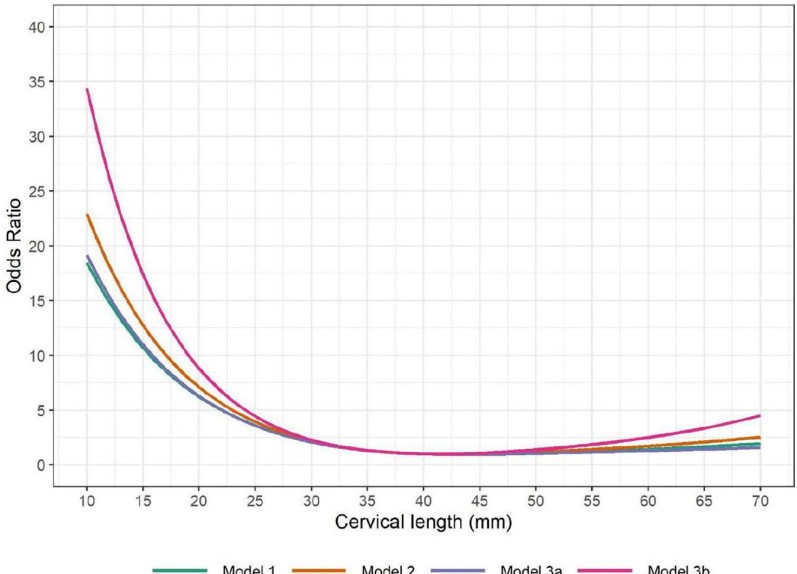

**Fig 4. Non-linear association between cervical length and spontaneous preterm birth <37 weeks in different models.** This figure illustrates ORs of spontaneous preterm birth <37 weeks at different cervical lengths vs. the reference cervical length (40 mm) in three models: Model 1: including only cervical length and gestational age at measurement as prognostic factors; adjusted for gestational age at measurement only. Model 2: including only key demographic factors (mother's age, nulliparity), cervical length, and gestational age at measurement. Adjusted for gestational age at measurement, maternal age, nulliparity. Model 3: including all potential available prognostic factors (Model 3a). Adjusted for gestational age at measurement, maternal age, maternal BMI, nulliparity AND/OR history of preterm birth. If any of these factors were missing within a study, then they were not adjusted for in the first stage for that particular study, but included in the meta-analysis at the second stage. A model with all potential prognostic factors, only including studies that include prognostic factors (Model 3b). Adjusted for gestational age at measurement, maternal age, maternal BMI, nulliparity AND history of preterm birth. If any of these factors were missing within a study, then that study was excluded from the meta-analysis in the second stage.

cervical length as a categorical prognostic factor (six used 25 mm as the cutoff) and only three analysed cervical length as a continuous prognostic factor. None considered non-linear associations between cervical length and SPTB in their analyses. Dunn and colleagues reported an OR of 0.92 (95% CI [0.87,0.98]) per 1 mm increase in cervical length [47]. Movahedi and colleagues reported an adjusted OR of 0.82 (95% CI [0.75.0.90]) per 1 mm increase in cervical length [48]. Wikstrom and colleagues reported adjusted ORs per 5 mm decrease in cervical length for three risk groups: high-risk women (OR 1.32; 95% CI [1.07,1.63]), nulliparous with no risk factors (OR 1.69; 95% CI [1.45,1.98]), and parous women with no risk factors (OR 1.22; 95% CI [1.00,1.48]) [49]. Therefore, it would not be possible to perform meta-analyses based on aggregate data presented in these studies or to consider non-linear associations as we did in our IPDMA.

## Discussion

50.7% of IPD (91,204/179,886 singleton pregnancies) from 27 studies was obtained in this IPDMA. The primary model assessing the association between cervical length nearest to 20 weeks' gestation and odds of SPTB performed at least as well as multivariable models. It showed an L-shape non-linear association between cervical length and SPTB. A longer cervical length was associated with steeply lower odds of SPTB until it reached 40 mm, beyond which the curve of cervical length on SPTB <37 weeks became flat. Multiple sensitivity analyses confirm the robustness of the findings. Subgroup analyses showed that the prognostic value of cervical length was stronger in women with a history of cervical surgery than those without cervical surgery; and was stronger in women without a history of preterm birth than those with a previous preterm birth.

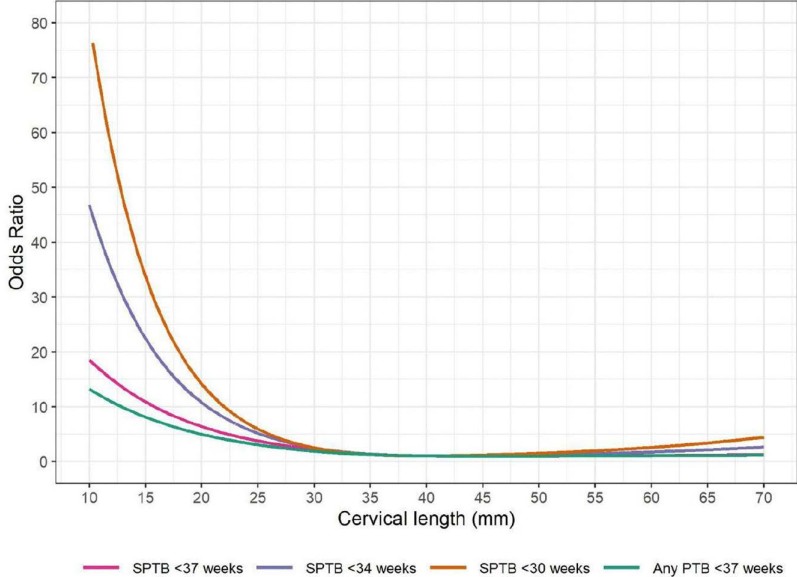

**Fig 5. Non-linear association between cervical length and different outcomes.** This figure illustrates the ORs of spontaneous preterm birth <37 weeks, <34 weeks, <30 weeks, and any preterm birth <37 weeks at different cervical length vs. the reference cervical length (40 mm) in the primary model. Cervical length and gestational age at measurement were the only prognostic factors used in the model.

Strengths of this meta-analysis include a comprehensive literature review designed in consultation with an information specialist, and extensive attempts to obtain IPD, including from grey literature. We have treated cervical length as a prognostic factor (rather than as a diagnostic test as in most previous literature) [8] and utilised statistical methods for prognostic factor research. We have avoided data loss associated with dichotomisation where possible [50,51] by analysing cervical length and GA as continuous variables and explored non-linear associations by using restricted cubic splines [18]. We have adhered to recommended reporting guidelines (S1 Checklist).

The null findings of our subgroup analysis of women with congenital uterine anomalies add weight to previous studies which have suggested that this group are at an increased risk of SPTB; this appears to occur independently of a short cervix [52,53]. Therefore, cervical length surveillance is unlikely to have more benefits for women with congenital uterine anomalies compared to those without, however they should continue to be treated as a group at high risk of spontaneous preterm birth. Of particular clinical relevance also is that for women with a history of preterm birth, at the prognostic value of cervical length for SPTB appears weaker in those with a history of preterm birth than those without, at shorter cervical length <25 mm. The reverse is true for women with a history of cervical surgery—the prognostic value of cervical length for SPTB appears stronger among them compared to those without a history of cervical surgery, at shorter cervical length <28 mm (<28 mm). We would encourage clinicians to give weight to all three risk factors when assessing a woman's risk of SPTB.

A limitation of our analysis is the proportion of IPD that was unavailable (approximately 50% of potential participants). Thirty-eight unavailable studies were due to a lack of author contact details, or because authors did not respond. Earlier evidence showed less than half of IPDMAs of randomized trials published before 2015 had >80% IPD retrieval rate [54], and this figure does not seem to have improved over time [55]. Thirteen percent of IPDMAs of randomized trials could not obtain data from >50% of included studies [55]. It is expected that IPDMAs of observational studies have lower retrieval rates, as data sharing for observational studies is less standardised as compared to trials and may be more restricted due complexity in governance and privacy restrictions [56]. We found several authors had left the institutions at which they

had previously published, their contact details were no longer valid and/or they were unable to be found at other institutions. While our IPD retrieval rate is suboptimal, the large sample size (>90,000) represented a wide variety of nations and patient risk profiles, which improves the generalizability of results. In addition, other research in women's health showed better quality and trustworthiness of studies sharing IPD compared to those not sharing IPD [57,58]. Additionally, the data collection phase occurred during the early years of the COVID-19 pandemic, which meant many authors were restricted in their research activities or illness affected themselves or their families directly. Comparison between available and unavailable datasets is limited by reporting, as many unavailable datasets were only published as conference abstracts. Nevertheless, based on the available information provided, the retrieved IPD still represent the overall population (Tables B and D in S1 Appendix).

We have described the challenges of the often convoluted and lengthy data sharing processes in a related article [59]; this has led to a much longer-than-desired interval between conclusion of the literature search, closure of the dataset, and the completion of this study. Legal negotiations between institutions at times took years to complete. This is frustrating for investigators, means literature reviews become outdated, and delays the benefits of new research findings from reaching patients. Primary study enrolment and consent processes already protect participant privacy, and adding layers of additional requirements for data management and transfer is of unclear benefit to the participant and only increases the time taken for research advancements to benefit the wider community. Our experience was similar to other reports in the literature on the prolonged process of IPDMA due to challenges in communication, changes in national or institutional data sharing policies, administrative, statistical, and legal barriers [60]. Had we been able to obtain approximately half of the cases from the updated literature search, our combined dataset could have been up to 25% larger, however, this is unachievable within a short time frame. We would again urge institutions to consider streamlining approaches to data sharing, and that participants' consent to participate in research should also apply to the use of their data in meta-analyses.

We have also previously described the limitations in aggregate data meta-analyses addressing the association between mid-trimester transvaginal cervical length and SPTB [8]. We have attempted to address these issues with our IPD-MA, however, many more articles were published after the closure of data transfer process. Performing an updated search revealed 26 more studies that could potentially be suitable for inclusion in an IPD-MA, although only three of them reported cervical length as a continuous prognostic factor. The issues that precluded pooling of data in previous meta-analyses remain in these newly eligible primary studies: inappropriate use of diagnostic test accuracy methods to address a prognostic factor research question and dichotomization of continuous variables with varied cutoffs. The three studies from our updated search that analyzed continuous variables did not consider non-linearity, although all three studies supported our findings that mid-trimester cervical length is an important prognostic factor for SPTB.

Risk of bias was quite high overall, and this was often due to study design. Retrospective cohort studies are inherently at high risk of bias in the participation domain of QUIPS (Domain 1) as participants are selected due to complete predictor and outcome data. While this design allows convenient data collection, it allows for no attempts to collect data from participants lost to follow-up. We have previously provided a potential example of an at-risk group of women who could be less likely to deliver at their booked hospital [59]. Another group of women excluded from this study design is those who have an iatrogenic PTB. Our models showed a similar predictive capacity of cervical length for both SPTB and all PTB, however, retrospective cohort studies almost invariably exclude women with iatrogenic PTB, which would likely influence this result. We would suggest that future studies do not exclude these women, but that iatrogenic preterm birth be treated as a competing event, and that multiple imputation be used for missing data.

We intended to include a variety of potential co-predictors along with cervical length in our analysis, and to investigate neonatal outcomes as a parallel project. Unfortunately, most datasets lacked information on these potential co-predictors, which limited the scope of the analysis, even for well-established factors such as smoking. However, we did ascertain that adding BMI and nulliparity to the model showed similar prognostic values of cervical length on SPTB. Data on neonatal outcomes were rarely collected, so this avenue could not be explored.

This analysis supports a non-linear association between cervical length and hazard of SPTB, with a steeper increase in incidence of SPTB as cervical length decreases below 40 mm. However, where cervical length is beyond this limit, it appears to have minimal to no association with risk of SPTB, perhaps because preterm births occurring in this subgroup may be due to other underlying mechanisms (e.g., placental abruption).

It is important to note that women with a short cervix (e.g., 10 mm) deliver after 37 weeks at a much lower frequency than those with a longer cervix, so we anticipate these risk estimates are less precise due to a smaller sub-population size.

We have performed a similar IPDMA in women with twin pregnancy [59], and in contrast to this study, the relationship between cervical length and length of gestation was confirmed to be linear. The IPDMA on twin pregnancies was based on IPD of 6,437 women from 17 studies while the IPDMA on singleton pregnancies was based on IPD of 91,204 from 27 studies. The IPDMA on twin pregnancies showed a slightly shorter cervical length compared to our current IPDMA on singleton pregnancies (twin versus singleton pregnancies (39±9 mm versus 40±9 mm), but much higher rates of SPTB <37 weeks (44.9% versus 5.1%) and SPTB <34 weeks (14.9% versus 1.6%). Such a large difference in SPTB can be expected, as twin pregnancy itself is a strong risk factor for SPTB. In the twin IPDMA, Cox regression analysis showed that for every additional 1 mm of cervical length, the rate of SPTB before 34 weeks decreased by 6.8% (hazard ratio [HR] 0.93 [0.92, 0.95]), and SPTB before 37 weeks decreased by 4.0% (HR 0.96 [0.95, 0.97]). The prognostic value remains consistent across the spectrum of cervical length distributions in the twin pregnancy IPDMA, including those with cervical length greater than 40 mm. The findings were consistent in logistic regression analysis. By contrast, we did not observe a consistent prognostic value in the singleton pregnancy IPDMA, as the prognostic value becomes very limited when cervical length is above 40 mm. We hypothesize that twin pregnancy itself is a such a strong risk factor for SPTB, with uterine overdistension likely a major contributing mechanism [2]. Therefore, in twin pregnancies, even when cervical length is over 40 mm, longer cervix would still be protective for SPTB despite the higher baseline SPTB rate among in this population.

The statistical approach we have used in this analysis is in keeping with recommendations for prognostic factor research, which can serve as the first step for more sophisticated development and validation of prediction models. More detailed and intentionally planned data collection including co-predictors will allow the development of prediction models to predict SPTB and subsequent external validation studies. These are beyond the scope of this manuscript, but could be a natural extension. A desirable outcome of additional work would include the ability to calculate SPTB risk for an individual, and to integrate this with the risk reduction offered by prophylactic treatment. An appropriately validated prognostic model would allow this type of individual risk assessment.

In conclusion, we found a non-linear association between cervical length and spontaneous preterm birth. Shorter cervix is associated with progressively higher risk of SPTB when cervical length is under 40 mm, but the probability of term birth is high when cervical length is over 40 mm. We recommend expected treatment effect size (including risks) be considered when selecting a threshold at which to offer prophylactic treatment. The interactions between historical risk factors (cervical surgery, previous preterm birth) at shorter cervical lengths can be further investigated with prediction modelling and used to guide personalised clinical management.

## Supporting information

**S1 Appendix. Table A:** Search strategy. Terms used in literature searches. **Table B:** Characteristics of studies without IPD. Studies for which IPD could not be obtained are listed, along with brief descriptors to allow comparison with the studies included in the analysis. **Table C:** Statistics of all analyses on non-linear associations. Results of cubic spline analyses are presented in this table. **Table D:** Characteristics of new studies identified during updated literature search. **Table E:** Contact details of authors of studies with IPD. **Fig A:** Scatter plot of cervical length (mm) vs. gestational age at birth (weeks). **Fig B1–B4:** Non-linear associations between cervical length and primary outcome in different models. **Fig

**C1–C3:** Non-linear associations between cervical length and secondary outcomes. **Fig D1–D5:** Sensitivity analyses for the primary outcome. **Fig E1–E6:** Subgroup analyses for the primary outcome.
(DOCX)

**S1 Code. Coding for statistical analyses is provided.**
(RMD)

**S1 Checklist. These checklists show adherence to PRISMA reporting guidelines.** PRISMA 2020 for Abstracts Checklist. Available from https://www.prisma-statement.org/prisma-2020-flow-diagram. This work is licensed under CC BY 4.0. To view a copy of this license, visit https://creativecommons.org/licenses/by/4.0/. Page MJ, McKenzie JE, Bossuyt PM, Boutron I, Hoffmann TC, Mulrow CD, et al. The PRISMA 2020 statement: an updated guideline for reporting systematic reviews. BMJ. 2021;372:n71. https://doi.org/10.1136/bmj.n71.
(DOCX)

**S2 Checklist. PRISMA 2020 abstract checklist.** PRISMA 2020 checklist. Available from https://www.prisma-statement.org/prisma-2020-flow-diagram. This work is licensed under CC BY 4.0. To view a copy of this license, visit https://creativecommons.org/licenses/by/4.0/. Page MJ, McKenzie JE, Bossuyt PM, Boutron I, Hoffmann TC, Mulrow CD, et al. The PRISMA 2020 statement: an updated guideline for reporting systematic reviews. BMJ. 2021;372:n71. https://doi.org/10.1136/bmj.n71.
(DOCX)

## Acknowledgments

We would like to acknowledge and thank Anne Young, clinical librarian at Monash University for her help with our literature search.

## Author contributions

**Conceptualization:** Kelly Hughes, Anna Lene Seidler, Ben Willem Mol, Rui Wang.

**Data curation:** Mason Aberoumand, Heather Ford, Erin Clarke, Megan Hall.

**Formal analysis:** David Nguyen, Mason Aberoumand, Anna Lene Seidler, Rui Wang.

**Funding acquisition:** Ben Willem Mol, Rui Wang.

**Investigation:** Kelly Hughes, Mason Aberoumand, Heather Ford, Erin Clarke, Rui Wang.

**Methodology:** Kelly Hughes, Mason Aberoumand, Pihla Kuusela, Anna Lene Seidler, Ben Willem Mol, Rui Wang.

**Project administration:** Kelly Hughes, Mason Aberoumand.

**Resources:** Nuria Banos Lopez, Margaret Dziadosz, Richard Fischer, Renato T. Souza, Jose Guilherme Cecatti, Kelly Orzechowski, Courtney Olson-Chen, Alberto Borges Peixoto, Vorapong Phupong, Joshua Rosenbloom, Moeun Son, Athena Souka, Liu Du, Michael Sean Esplin, Roberta Granese, Simi Gupta, Brenda Kazemier, Lindsay Kindinger, Pihla Kuusela, Jeanine Van der Ven, Omer Weitzner, Evelyn Minis, Alba Farras Llobet, Heather Frey, Rashmi Bagga, Siddhidatri Mishra, Elizabeth Patberg, Philip Bennett, Megan Hall, Andrew Shennan, Ben Willem Mol.

**Software:** Mason Aberoumand.

**Supervision:** Shaun Brennecke, Shakila Thangaratinam, Anna Lene Seidler, Ben Willem Mol, Rui Wang.

**Visualization:** Mason Aberoumand.

**Writing – original draft:** Kelly Hughes.

**Writing – review & editing:** Kelly Hughes, David Nguyen, Mason Aberoumand, Heather Ford, Erin Clarke, Nuria Banos Lopez, Margaret Dziadosz, Richard Fischer, Renato T. Souza, Jose Guilherme Cecatti, Kelly Orzechowski, Courtney Olson-Chen, Alberto Borges Peixoto, Vorapong Phupong, Joshua Rosenbloom, Moeun Son, Athena Souka, Liu Du, Michael Sean Esplin, Roberta Granese, Simi Gupta, Brenda Kazemier, Lindsay Kindinger, Pihla Kuusela, Jeanine Van der Ven, Omer Weitzner, Evelyn Minis, Alba Farras Llobet, Heather Frey, Rashmi Bagga, Siddhidatri Mishra, Elizabeth Patberg, Philip Bennett, Megan Hall, Andrew Shennan, Shaun Brennecke, Shakila Thangaratinam, Anna Lene Seidler, Ben Willem Mol, Rui Wang.

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
