## [Editor Report · Decision Letter 0]

3 Jun 2025

Dear Dr Hughes,

Thank you for submitting your manuscript entitled "Prognostic value of cervical length for spontaneous preterm birth in asymptomatic women with singleton pregnancy: an individual participant data meta-analysis" for consideration by PLOS Medicine.

Your manuscript has now been evaluated by the PLOS Medicine editorial staff and I am writing to let you know that we would like to send your submission out for external peer review.

However, before we can send your manuscript to reviewers, we need you to complete your submission by providing the metadata that is required for full assessment. To this end, please login to Editorial Manager where you will find the paper in the ’Submissions Needing Revisions’ folder on your homepage. Please click ’Revise Submission’ from the Action Links and complete all additional questions in the submission questionnaire.

For clinical studies, please upload a copy of your trial study protocol as a supporting information file. The study protocol should be the version submitted for approval to the institutional review board or ethics committee, should include any amendments to the study protocol, as well as the date of their approval by the institutional review or ethics committee. Please also detail any deviations from the study protocol in the Methods section of your manuscript. The editors will consider the protocol and study conduct prior to a final decision for external review.

Please re-submit your manuscript within two working days, i.e. by Jun 05 2025 11:59PM.

Kind regards,

Louise Gaynor-Brook, MBBS PhD

Senior Editor

PLOS Medicine

---

## [Decision Letter · Decision Letter 1]

18 Sep 2025

Dear Dr Hughes,

Many thanks for submitting your manuscript "Prognostic value of cervical length for spontaneous preterm birth in asymptomatic women with singleton pregnancy: an individual participant data meta-analysis" (PMEDICINE-D-25-01857R1) to PLOS Medicine. The paper has been reviewed by subject experts and a statistician; their comments are included below and can also be accessed here: [LINK]

As you will see, the reviewers raise several concerns that will require thoughtful attention. After discussing the paper with the editorial team and an academic editor with relevant expertise, I’m pleased to invite you to revise the paper in response to the reviewers’ comments. We plan to send the revised paper to some or all of the original reviewers, and we cannot provide any guarantees at this stage regarding publication.

We ask that you submit your revision by Oct 09 2025. However, if this deadline is not feasible, please contact me by email, and we can discuss a suitable alternative.

Don’t hesitate to contact me directly with any questions (atosun@plos.org).

Best regards,

Alexandra

Alexandra Tosun, PhD

Senior Editor

PLOS Medicine

atosun@plos.org

Comments from the reviewers:

Reviewer #1: Alex McConnachie, Statistical Review

This review looks at the use of statistics in the paper by Hughes et al. This is an IPD meta-analysis of the association between cervical length measured at around 20 weeks’ gestation and the incidence of spontaneous preterm birth.

My overall impression is that the paper is unfinished. The version I have for review retains some comments between the authors, and the tables and figures are not clearly labelled. The assessment of linearity of the association of interest, described in the methods section, does not appear to be reported, though the abstract concludes by saying that the association is linear. The sensitivity analyses using logistic regression are reported (there are three forest plots titled "Model 2: SPTB < 37 weeks", "Model 3: SPTB < 34 weeks", and "Model 4: SPTB < 30 weeks", which I assume are the logistic regression models, since they report odds ratios), but do not seem to be commented on.

That being said, I think the analyses are on the right track.

Use of Cox models to look at the association between CL and the risk of SPTB is reasonable, but is complicated by the varying GA at measurement of CL. The authors deal with this problem by including GA at measurement as a covariate in the models (I wonder what these coefficients’ estimates were, but maybe that is just me?), but I would have been tempted to build the model using GA as the time variable. That way, the risk set over time would always consist of pregnancies at the same GA; the baseline hazard function would be modelling the risk of SPTB at each GA; it seems a model built that way would make more biological sense.

In addition, I would consider looking at whether there is an interaction between GA at measurement and the association between CL and SPTB. Other sensitivity analyses could look the association when the GA at measurement is restricted to tighter windows around the 20-week target.

Currently, there is no mention of assessing the proportional hazards assumption, but from the results presented, it appears that this assumption will not be met. The HR per mm increase in CL for SPTB up to 37 weeks is 0.93, but for SPTB up to 34 weeks is 0.90 (note, in the results section this is incorrectly reported as 0.93). These represent "average" hazard ratios over the whole follow-up period in each model. However, if the average HR up to 37 weeks is 0.93, but the average up to 34 weeks is 0.90, then the HR for SPTB between 34 and 37 weeks must be >0.93, suggesting that the HR is not constant over time.

Some other, non-PH model could perhaps be used, but this is at the edge of my knowledge, and I wonder whether it would be necessary to employ a one-step approach to the analysis to fit a more complex model. A simpler approach, and one more amenable to two-stage MA, might be a piecewise constant HR Cox model, and this may be easier to do if using GA as the time variable in the analysis.

Note, none of these issues are a problem if using logistic regression for the binary outcomes of SPTB <37, 34, or 30 weeks, and it is comforting to see that the results of these analyses are consistent with the others. The paper might be more accessible if using this approach as the primary method, though I would still want to consider whether the association between CL and SPTB varies according to the GA at which CL was measured.

Other things I noted:

In the abstract, the primary outcome is described as SPTB <37 weeks, but the analysis is a Cox model, so the primary outcome is actually the time (from CL measurement) to SPTB.

The conclusion of the abstract begins with "Cervical length has a linear, positive association with length of gestation." This is suggested by Supplementary Figure 2, though the linearity of this association is no assessed. However, the primary analysis looks at the (assumed linear) association between CL at around 20 weeks’ GA and the subsequent (log) hazard of SPTB up to 37 weeks; this is not the same thing as the first sentence of the abstract conclusions.

The next sentence in the abstract conclusion is "A single cervical length value cannot accurately exclude SPTB risk in all women", though there is nothing in the abstract, nor indeed in the paper, that specifically addresses this point (e.g. no table showing sensitivity, specificity, etc., of alternative cut-offs of CL). Whilst I believe the statement to be true, given the data presented, it needs to be more explicitly shown.

The subgroup analyses are performed by fitting interaction terms into the main model within each study, and meta-analysing these interaction terms. This is the correct approach, but for presentation, I am not interested in the pooled estimate of the interaction HR. As a reader all I want to know is the HR within each subgroup, and a p-value to tell me whether these are the same or different. A forest plot showing the interactions that were assessed would be welcome.

The issue of the four studies that were "accidentally omitted from the analysis due to an administrative error" is mildly troubling. If it was just an administrative error, why can that not be corrected and these studies included? I applaud the authors for their honesty here, but I want to know why these studies can never be included.

I don’t see how reporting prediction intervals for the pooled HRs is useful.

In summary, I like the analyses that are being done, and I like the paper as a whole (the descriptions of the data sources and exposures and outcomes were all quite good, I thought), but I feel the analysis and reporting of results (tables and figures in particular) need to be tidied up a little. Personally, I would either do the Cox analysis differently (with GA as the time variable, though the non-PH needs to be dealt with) or use logistic regression for the binary outcomes, but either way I would like to know whether the association depends on when CL was measured.

Reviewer #2: Summary of the study

This is an individual participant data (IPD) meta-analysis evaluating the prognostic value of mid-trimester transvaginal cervical length (CL) for spontaneous preterm birth (SPTB) in asymptomatic singleton pregnancies. The authors assembled IPD from large cohorts and trials (reported ~88,498 pregnancies), modeled CL as a continuous predictor with time-to-event outcomes (<37 and <34 weeks) using two-stage Cox models, explored non-linearity with penalized splines, and examined prespecified subgroups (e.g., prior PTB, cervical surgery, parity). They conclude that every 1-mm increase in CL is associated with a ~7% lower hazard of SPTB <37w and ~10% lower hazard <34w; effect sizes appear stronger in women with prior cervical surgery, in nulliparas, and in those without a prior PTB. The manuscript argues against a single "critical" CL cut-off and provides a table translating CL values to cumulative risks for clinical counseling.

Comments

1. Numerical and internal inconsistencies

o The manuscript reports different events across sections (e.g., "SPTB events <37w reported as 4,276 in the Abstract vs 4,356 in Results). Please reconcile all numbers (records/studies/participants/events) consistently across Abstract, Methods, Results, PRISMA, and tables.

o PROSPERO registration appears inconsistent (CRD42019137400 in Abstract vs CRD42020146987 in Methods). Clarify which is correct and ensure all occurrences match.

o Several typos in affiliations and names (e.g., "Chulalongkom" vs "Chulalongkorn"; "Merck Merck KGaA").

2. IPD yield, and risk of bias

o Only ~44% of potential IPD were obtained, with >50% unavailable. This raises concerns about selection bias at the study level. Please quantify and compare characteristics (publication year, geography, sample size, risk profile, CL distribution, outcome rates) between included vs unavailable datasets to assess representativeness.

3. Clinical translation

o For clinical application, please consider providing the following:

a) A calculator/nomogram (code and formula) so clinicians can compute risks for their population;

b) External validation or, at minimum, study-level calibration plots to show portability;

c) A brief decision-analytic scenario (e.g., thresholds for offering progesterone/cerclage) with net-benefit analysis.

Reviewer #3: The intention of the article is to be congratulated. However, even though the aim of an IPD is to reduce bias due to small sample sizes, the risk of selection bias remains considerate. Iatrogenic preterm births are handled as censoring events, which, in my opinion, is not correct, since they do not represent patients that have the same profile than those who remain follow-up.

Abstract

"We searched Medline, Embase, Cochrane, and LILACS to identify studies which could

assess association between mid-trimester transvaginal sonographic cervical length

and SPTB in asymptomatic women with singleton pregnancy" => "which could assess an (or the) association"

"occurred in 4276 (5.1)%" => "occurred in 4276 (5.1%)"

Introduction

"however spontaneous preterm birth (SPTB) represents a potentially predictable and preventable condition" => why do you use "however" in this sentence? iatrogenic preterm birth also is a possible predictable and preventable condition, for example iatrogenic PTB due to placental insufficiëncy

Methodology

"We searched Medline, Embase, CINAHL, LILACS, Database of Abstracts of Reviews of Effects (DARE), Cochrane database, JBI Database of Systematic Reviews, ClinicalTrials.gov and Google Scholar." => in the abstract less databases are mentioned

"Where studies also reported medically indicated preterm birth, these were treated as censored observations." =>

- so if studies did not include iatrogenic PTBs, these were not included? this means you have a mix of populations, which can bias the result

- iatrogenic preterm birth is not an event that requires censoring, it is a competing event: when iatrogenic preterm birth occurs, sPTB cannot occur anymore. By censoring preterm birth you hypothesise that the patient is still pregnant, that she might still experience sPTB and that her profile does not differ from the patients that remain in your cohort (independent censoring)

- excluding iatrogenic preterm births results in a biased population (selection bias), which is not representative for the population of interest since at the time of measuring the cervix, the outcome of the patient is not know yet

"Complete case analysis was performed." => how many cases were incomplete?

Results

"IPD was sought for 109 eligible citations, however for 64 of them…" => add the % (58.7%); very high, non-reassuring, number which should be addressed in the discussion (how is it possible and what are the implications)

"four datasets (n=3993) were accidentally omitted from the analysis due to an administrative error" => is it too/that much work to rerun the analyses with this additional data?

"Planned subgroup analysis showed stronger prognostic value of cervical length on SPTB< 37 weeks in women with a history of cervical surgery (4 studies, n=36 416, interaction HR=0.96 (4 studies, n=36 416, interaction HR=0.96 (0.94-0.98), PrI 0.92 - 1.01, tau2 = 0). In women without a history of cervical surgery, each additional millimetre of cervical length was associated with a 7% (7-8%) decreased hazard of SPTB <37 weeks’ gestation, whereas those

who had previous cervical surgery, each additional millimetre of cervical length was associated with an 11% (9-12%) decreased hazard of SPTB <37 weeks." => the first sentence is not necessary, since the interaction term is significant, the cervical lengths should be considered within the subgroups (surgery vs no surgery); what is meant by surgery?

"(17 studies, n=82 168, interaction HR=0.97 (0.96 - 0.99), PrI 0.93-1.02, tau2 = 0.0003." => close the brackets

Discussion

"10% reduction in risk of SPTB prior to 34 weeks’ gestation for every millimetre increase in

cervical length and a 7% reduction in risk" => it is not a reduction in risk but in hazard; this should also be taken into consideration for the text that follows

"Subgroup analyses showed a stronger prognostic value of cervical length on SPTB< 37 weeks in nulliparous women, women with a history of cervical surgery, and parous women without a previous preterm birth." => which explanations do you hypothesize?

"Of particular clinical relevance also is that cervical length has less prognostic value for SPTB in women who have had a previous spontaneous preterm birth" => were prophylactic measures taken in the majority of these patients?

" The possibility of systematic exclusion of a group of women with similar characteristics cannot be dismissed." => if the patients have the same characteristics, that is not such a problem, it is a problem if the excluded patients differ from the included ones (but are part of the population screened at 20 weeks)

"We would suggest that future studies do not exclude these women, but that censoring is used where PTB is indicated for maternal or fetal reasons, and to use multiple imputation for missing data." => as mentioned, censoring iatrogenic birth is not the correct way to handle competing events

"However, where cervical length is at the upper limit, it appears to have minimal to no impact on the risk of preterm birth, probably because preterm births occurring in this subgroup would be due to other underlying mechanisms." => what is the rationale to write this? do you mean other underlying mechanisms leading to spontaneous preterm birth? or other causes of PTB other than sPTB/PPROM? if you mean the latter, weren’t these PTBs excluded/censored?

"It is important to note that women with a short cervix (e.g. 10mm) deliver after 37 weeks at a much lower frequency than those with cervical length toward the middle of the bell curve, so we anticipate these risk estimates are less precise." => I don’t understand what you mean by this sentence

"Transvaginal cervical length measured in the second trimester in asymptomatic women with singleton pregnancy is associated with risk of spontaneous preterm birth…" => the way this is written, you suggest that cervical length measurement is associated with sPTB, it is not the measurement an sich, but the result of the measurement: the shorter the cervix, the higher the risk, without an evident cut-off (and the best cut-off could be explored using a ROC curve, without forgetting to look at calibration)

"Number of eligible records for which IPD were not sought: 2 records, n=7935" => why were they not sought?

In the discussion, could you elaborate on the (implications of) the varying sPTB incidences in the included cohorts.

Figures/tables

Suppl figure 1: are all the orange dots censored patients? (in your case iatrogenic births)

---

* Please upload any figures associated with your paper as individual TIF or EPS files with 300dpi resolution at resubmission; please read our figure guidelines for more information on our requirements: http://journals.plos.org/plosmedicine/s/figures. While revising your submission, we strongly recommend that you use PLOS’s NAAS tool (https://ngplosjournals.pagemajik.ai/artanalysis) to test your figure files. NAAS can convert your figure files to the TIFF file type and meet basic requirements (such as print size, resolution), or provide you with a report on issues that do not meet our requirements and that NAAS cannot fix.

After uploading your figures to PLOS’s NAAS tool - https://ngplosjournals.pagemajik.ai/artanalysis, NAAS will process the files provided and display the results in the "Uploaded Files" section of the page as the processing is complete.

If the uploaded figures meet our requirements (or NAAS is able to fix the files to meet our requirements), the figure will be marked as "fixed" above. If NAAS is unable to fix the files, a red "failed" label will appear above.

When NAAS has confirmed that the figure files meet our requirements, please download the file via the download option, and include these NAAS processed figure files when submitting your revised manuscript.

* Please ensure that the paper adheres to the PLOS Data Availability Policy (see http://journals.plos.org/plosmedicine/s/data-availability), which requires that all data underlying the study’s findings be provided in a repository or as Supporting Information. For data residing with a third party, authors are required to provide instructions with contact information (web or email address) for obtaining the data. Please note that a study author cannot be the contact person for the data. PLOS journals do not allow statements supported by "data not shown" or "unpublished results." For such statements, authors must provide supporting data or cite public sources that include it.

* We expect all researchers with submissions to PLOS in which author-generated code underpins the findings in the manuscript to make all author-generated code available without restrictions upon publication of the work. In cases where code is central to the manuscript, we may require the code to be made available as a condition of publication. Authors are responsible for ensuring that the code is reusable and well documented. Please make any custom code available, either as part of your data deposition or as a supplementary file. Please add a sentence to your data availability statement regarding any code used in the study, e.g. "The code used in the analysis is available from Github [URL] and archived in Zenodo [DOI link]" Please review our guidelines at https://journals.plos.org/plosmedicine/s/materials-software-and-code-sharing and ensure that your code is shared in a way that follows best practice and facilitates reproducibility and reuse. Because Github depositions can be readily changed or deleted, we encourage you to make a permanent DOI’d copy (e.g. in Zenodo) and provide the URL.

* Please add this statement to the manuscript’s Competing Interests: "AS is an Academic Editor on PLOS Medicine’s editorial board."

* The funding statement should include: specific grant numbers, initials of authors who received each award, URLs to sponsors’ websites. Also, please state whether any sponsors or funders (other than the named authors) played any role in study design, data collection and analysis, the decision to publish, or preparation of the manuscript. If they had no role in the research, include this sentence: “The funders had no role in study design, data collection and analysis, decision to publish, or preparation of the manuscript.”

* The Data Availability Statement (DAS) requires revision. Currently, it is unclear how readers interested in the data could contact the authors of the primary studies. Could you include a list of contacts, as was done in the article https://doi.org/10.1371/journal.pmed.1004502?

* At this stage, we ask that you include a short, non-technical Author Summary of your research to make findings accessible to a wide audience that includes both scientists and non-scientists. The Author Summary should immediately follow the Abstract in your revised manuscript. This text is subject to editorial change and should be distinct from the scientific abstract. Ideally each sub-heading should contain 2-3 single sentence, concise bullet points containing the most salient points from your study. In the final bullet point of ’What Do These Findings Mean?’, please include the main limitations of the study in non-technical language. Please see our author guidelines for more information: https://journals.plos.org/plosmedicine/s/revising-your-manuscript#loc-author-summary.

FIGURES AND TABLES

SUPPLEMENTARY MATERIAL

REFERENCES

STUDY TYPE-SPECIFIC REQUESTS

* Please report your SR/MA according to the PRISMA guidelines provided at the EQUATOR site. http://www.equator-network.org/reporting-guidelines/prisma/. Please provide the completed PRISMA checklist as Supporting Information. When completing the checklist, please use section and paragraph numbers, rather than page numbers. Please add the following statement, or similar, to the Methods: "This study is reported as per the Preferred Reporting Items for Systematic Reviews and Meta-Analyses (PRISMA) guideline (S1 Checklist)."

* Abstract: Please report your abstract according to PRISMA for abstracts (https://doi.org/10.1371/journal.pmed.1001419) following the PLOS Medicine abstract structure (Background, Methods and Findings, Conclusions). Please ensure you provide dates of search, data sources, number of studies included, types of study designs included, eligibility criteria, and synthesis/appraisal methods.

* Please note that we expect searches to be updated to within 6 months of the time of submission.

* Please note that if you are unable to update the search due to the IPD nature of your study, we at minimum expect you to rerun the search and report how many studies (and pregnancies) would have qualified for inclusion. Please also include a discussion about how these studies could have influenced the results.

---

## [Decision Letter · Decision Letter 2]

15 Jan 2026

Dear Dr Hughes,

Many thanks for re-submitting your manuscript "Prognostic value of cervical length for spontaneous preterm birth in asymptomatic women with singleton pregnancy: an individual participant data meta-analysis" (PMEDICINE-D-25-01857R2) to PLOS Medicine. The paper has been seen again by two of the original reviewers, including a statistician; their comments are included below and can also be accessed here: [LINK]

As you will see, the reviewers are mostly satisfied with the revised manuscript. Since Reviewer #3 had several concerns about the statistics, we consulted with a statistical reviewer. Their feedback is included below. Reviewer #3 also raised a concern about the IPD-MA being considerably outdated, which sparked a discussion among the editorial team. We are aware that our minimum requirement was to rerun the search, report how many studies (and pregnancies) would have qualified for inclusion, and include a discussion about how these studies could have influenced the results. We don’t believe the influence of these manuscripts has been sufficiently discussed/addressed, especially given the large number of potential manuscripts to be included. After further consideration, we have decided that rerunning the search is insufficient in this case. For us to further consider your manuscript, please conduct a standard study-level meta-analysis of the 41 studies to see how the results compare. Additionally, we ask that you compare the findings of your current study in more detail with those of the IPD meta-analysis on twin pregnancies that you conducted. We appreciate that this is a challenging task, but we believe it will maximize the impact of your work.

Please revise the paper in response to the reviewers’ and editors’ comments. We plan to send the revised paper to some or all of the original reviewers, and we cannot provide any guarantees at this stage regarding publication.

We ask that you submit your revision by Feb 05 2026. If this deadline is not feasible in light of our requests, please contact me by email and we can discuss a suitable alternative.

Don’t hesitate to contact me directly with any questions (atosun@plos.org).

Best regards,

Alexandra

Alexandra Tosun, PhD

Senior Editor

PLOS Medicine

atosun@plos.org

Comments from the reviewers:

Reviewer #1: Alex McConnachie, Statistical Review

I thank the authors for their consideration of my original comments, and the work they have done to address them. They have done everything I asked, and I am happy with the changes they have made to the paper.

For me, the paper tells a good story, with quite a clear message. I have no further comments to make.

Reviewer #2: see attachment

Additional feedback from the statistical reviewer:

1) "Answer directed to reviewer #1, question 2:..." - The statistical reviewer finds that what you have done with the Cox models is in line with what they had suggested. You used GA as the time variable and said that each pregnancy was "under observation" from GA at measurement to GA at delivery.

2) "Answer directed to reviewer #1, question 3:..." - The statistical reviewer finds this acceptable. You are using the logistic regression model as the main analysis and the Cox model as a sensitivity analysis. Even if there is non-PH, one can always view the HR as an average across the time frame. Checking the PH assumption would be more important if the Cox model were being retained as the main analysis.

3) "Addition to question 5 of reviewer #1:..." - The statistical reviewer points out that this was included in the revised analysis (see response to comment 2) and that no evidence of an interaction was found.

---

## [Decision Letter · Decision Letter 3]

17 Mar 2026

Dear Dr. Hughes,

Thank you very much for re-submitting your manuscript "Prognostic value of cervical length for spontaneous preterm birth in asymptomatic women with singleton pregnancy: an individual participant data meta-analysis" (PMEDICINE-D-25-01857R3) for review by PLOS Medicine.

Thank you for your detailed response to the reviewers’ and editors’ comments. I have discussed the paper with my colleagues, and it has also been seen again by two of the original reviewers. The changes made to the paper were satisfactory to the reviewers. As such, we intend to accept the paper for publication, pending your attention to the reviewers’ and editors’ comments below in a further revision. When submitting your revised paper, please once again include a detailed point-by-point response to the editorial comments. The remaining issues that need to be addressed are listed at the end of this email.

In revising the manuscript for further consideration here, please ensure you address the specific points made by each reviewer and the editors. In your rebuttal letter you should indicate your response to the reviewers’ and editors’ comments and the changes you have made in the manuscript. Please submit a clean version of the paper as the main article file. A version with changes marked must also be uploaded as a marked up manuscript file. Please also check the guidelines for revised papers at http://journals.plos.org/plosmedicine/s/revising-your-manuscript for any that apply to your paper.

Please note, when your manuscript is accepted, an uncorrected proof of your manuscript will be published online ahead of the final version, unless you’ve already opted out via the online submission form. If, for any reason, you do not want an earlier version of your manuscript published online or are unsure if you have already indicated as such, please let the journal staff know immediately at plosmedicine@plos.org.

We ask that you submit your revision by Mar 24 2026. However, if this deadline is not feasible, please contact me (atosun@plos.org) or the journal staff by email, and we can discuss a suitable alternative.

We look forward to receiving the revised manuscript.

Sincerely,

Alexandra Tosun, PhD

Senior Editor

PLOS Medicine

plosmedicine.org

Comments from Reviewers:

Reviewer #1: Alex McConnachie, Statistical Review

I am happy with the analyses; the authors seem to have dealt with all questions. I have no further comments.

Reviewer #3: Congratulations. Nice work!

Requests from Editors:

GENERAL

* Please confirm that your title complies with to PLOS Medicine’s style. Your title must be nondeclarative and not a question. It should begin with main concept if possible. "Effect of" (or “Impact of”) should be used only if causality can be inferred, i.e., for an RCT. Please place the study design ("A randomized controlled trial," "A retrospective study," "A modelling study," etc.) in the subtitle (ie, after a colon).

* Statistical reporting: Please revise throughout the manuscript, including tables and figures.

- Please report statistical information as follows to improve clarity for the reader, ""XX% (95% CI [XX,YY]; p</=)"".

- Please separate upper and lower bounds with commas instead of hyphens as the latter can be confused with reporting of negative values.

- Please repeat statistical definitions (HR, CI etc.) for each set of parentheses.

* Please ensure that all abbreviations are defined at first use throughout the text (including statistical abbreviations).

* Please ensure that tables and figures, including those in supplementary files, are appropriately referenced in the main text.

* Please review your text for claims of novelty or primacy (e.g. ’for the first time’ or ‘novel’) and remove this language.

* Please confirm that any use of statistical terms (such as trend or significant) are supported by the data, and if not please remove them. The term trend should be used only when the test for trend has been conducted.

* Please define all acronyms used in each figure or table in the corresponding legend.

* Please confirm that you used patient-centered language. Please note that patient-centered language is constructed with the use of post-modified nouns putting the person first in the sentence structure.

* Please include an ethics approval statement in the Methods section, including the approval number and consent details.

* Please include the statement on code availability in the data availability statement.

ABSTRACT

* Please confirm that your abstract complies with our requirements, including providing all the information relevant to this study type https://journals.plos.org/plosmedicine/s/submission-guidelines#loc-abstract

* Please confirm that all numbers presented in the abstract are present and identical to numbers presented in the main manuscript text.

* In the last sentence of the Abstract Methods and Findings section, please describe the main limitation(s) of the study’s methodology.

* Please name all nine databases.

* Please specify inclusion and exclusion criteria.

* Please summarize relevant characteristics of the studies included (e.g. cohort studies).

* Please quantify the main results (with 95% CIs).

AUTHOR SUMMARY

* In the author summary, in the final bullet point of ’What Do These Findings Mean?’, please include the main limitations of the study in non-technical language.

* “(e.g. before 3 4 weeks) with shorter cervix.” – please check (34 weeks?).

METHODS AND RESULTS

* Thank you for providing the completed PRISMA checklist as Supporting Information. Please replace the page numbers with paragraph numbers per section (e.g. "Methods, paragraph 1"), since the page numbers of the final published paper may be different from the page numbers in the current manuscript.

* “Identifying studies – search” – please include a reference to “Supplementary Table 1 Search strategy” here.

* Table 1: For the last two columns, please provide numerators and denominators. For ‘Mean GA at delivery’, please add ‘(SD)’ in the column header.

* Please note that the file you uploaded containing Table 2 does not contain the full table. Please check and revise Table 2 and the relevant main text for clarity:

a) “2738 women (3.0% of the cohort) received prophylactic treatment as shown in Table 2.” – Where is this information in Table 2?

b) Please clarify that gestational age is provided in weeks.

c) “4442 women (5.2%) had a SPTB <37 weeks’ gestation, and 1420 (1.6%) before 34 weeks.” – the numbers in the text do not match the numbers in the table. Please check.

d) “Treatment with history of preterm birth” – What are these numbers? Please provide statistical definitions.

e) What are the denominators for each group?

f) The column heading “Number of participants (%)/mean (SD)” does not appear suitable for all categories included in Table 2. Please check and revise.

* We consider the risk of bias assessment an integral component of any systematic review and meta-analysis. We suggest moving Supplementary Figure 2 into the main text.

* Supplementary Figures: Please explain why you provided an accompanying table for some, but not all, of the figures detailing the odds ratios according to cervical length. For example, Figure 4c contains a table, but Figures 4a and 4b do not. Please add a table like this to each relevant figure.

* Please confirm that where relevant figures include 95% CIs.

* Please confirm that you specified the variables controlled for in all relevant Tables.

* l.323, “3) Model with all potential prognostic factors (Model 3a), and model with all potential prognostic factors, only including studies that include all factors (Model 3b)” – We don’t think this description is sufficiently clear. Please revise for clarity, including the description of Figure 3.

DISCUSSION

* Please remove the ’conclusions’ subheading from the discussion. Please also remove any other subheadings from the discussion.

* “it appears to have minimal to no impact on the risk of preterm birth” - Your study is observational and therefore causality cannot be inferred. Please remove language that implies causality and refer to associations instead. Please revise throughout.

General Journal Requests

2) Please ensure that the paper adheres to the PLOS Data Availability Policy (see http://journals.plos.org/plosmedicine/s/data-availability), which requires that all data underlying the study’s findings be provided in a repository or as Supporting Information. For data residing with a third party, authors are required to provide instructions with contact information for obtaining the data. PLOS journals do not allow statements supported by "data not shown" or "unpublished results." For such statements, authors must provide supporting data or cite public sources that include it.

---

## [Editor Report · Decision Letter 4]

16 Apr 2026

Dear Dr Hughes,

On behalf of my colleagues and the Academic Editor, Annettee Nakimuli, I am pleased to inform you that we have agreed to publish your manuscript "Prognostic value of cervical length for spontaneous preterm birth in asymptomatic women with singleton pregnancy: an individual participant data meta-analysis" (PMEDICINE-D-25-01857R4) in PLOS Medicine.

I appreciate your thorough responses to the reviewers’ and editors’ comments throughout the editorial process. We look forward to publishing your manuscript, and editorially there is only one remaining point that should be addressed prior to publication. We will carefully check whether the change has been made. If you have any questions or concerns regarding these final requests, please feel free to contact me at atosun@plos.org.

Please see below the minor point that we request you respond to:

* Please change the Abstract conclusion to: "Shorter cervix is a potential prognostic factor for SPTB when cervical length is under 40mm, but its prognostic value appears limited when cervical length is over 40 mm."

Before your manuscript can be formally accepted you will need to complete some formatting changes, which you will receive in a follow up email (including the editorial request above). Please be aware that it may take several days for you to receive this email; during this time no action is required by you. Once you have received these formatting requests, please note that your manuscript will not be scheduled for publication until you have made the required changes.

PRESS

Sincerely,

Alexandra Tosun, PhD

Senior Editor

PLOS Medicine